# The Effect of Leisure-Time Exercise on Mental Health Among Adults: A Bibliometric Analysis of Randomized Controlled Trials

**DOI:** 10.3390/healthcare13050575

**Published:** 2025-03-06

**Authors:** Karuppasamy Govindasamy, Masilamani Elayaraja, Abderraouf Ben Abderrahman, Koulla Parpa, Borko Katanic, Urs Granacher

**Affiliations:** 1Department of Sports, Recreation and Wellness, Symbiosis International (Deemed University), Hyderabad Campus, Modallaguda (V), Nandigama (M), Rangareddy, Telangana 509217, India; 2Department of Physical Education and Sports, Pondicherry University, Puducherry 605014, India; elaya.cricket@pondiuni.ac.in; 3Higher Institute of Sport and Physical Education of Ksar-Said, University of Manouba, Tunis 1000, Tunisia; abderraouf.benabderrahman@issep.uma.tn; 4Tunisian Research Laboratory “Sports Performance Optimization” LR09SEP01, National Center of Medicine and Science in Sports (CNMSS), Tunis 1000, Tunisia; 5Faculty of Sport and Exercise Science, UCLan University of Cyprus, Pyla 7080, Cyprus; kparpa@uclan.ac.uk; 6Montenergin Sports Academy, 81000 Podgorica, Montenegro; borkokatanic@gmail.com; 7Department of Sport and Sport Science, Exercise and Human Movement Science, University of Freiburg, 79102 Freiburg, Germany

**Keywords:** bibliometric, depression, leisure time, mental health, physical activity, sustainable cities

## Abstract

**Background:** Adequate levels of leisure-time exercise (LTE) are associated with mental health benefits. Despite increased research in recent years through randomized controlled trials (RCTs), a systematic literature review summarizing these findings is lacking. Here, we examined publication trends, impact, and research gaps regarding LTE’s effects on mental health in the form of a bibliometric analysis. **Methods:** Five electronic databases (PubMed, EMBASE, Web of Science, Ovid Medline, and the Cumulative Index for Nursing and Allied Health Literature) were searched from their inception until 20 November 2024. Citations were independently screened by two authors and included based on pre-determined eligibility criteria. Bibliometric analysis was conducted using SciVal and VOSviewer under five themes: (1) descriptive analysis, (2) network analysis, (3) thematic mapping, (4) co-citation and co-occurrence analysis, and (5) bibliometric coupling. **Results:** The systematic search identified 5792 citations, of which 78 RCTs met the inclusion criteria. Only one study was conducted in a low- or middle-income country. Sixty-four percent of studies were published in quartile-one journals. Most studies were conducted in the United States, followed by Australia, Canada, and the United Kingdom. National collaborations yielded the highest citation rates, reflecting the influence of cultural and social norms on exercise and mental health. Research gaps were identified with regards to the validity of mental health measures, the paucity of data from low- and middle-income countries, and emerging research sources. **Conclusions:** This bibliometric analysis highlights the existing evidence on LTE’s impact on mental health and identifies areas for future research and policy. Trials exploring valid mental health outcomes, biomarkers such as mood and oxidative stress, and collaborative research are needed, particularly in underrepresented regions of the world.

## 1. Introduction

Mental health, a state of well-being that enables individuals to cope with daily stressors, has been a vital component of social policy and human rights [1]. The Global Burden of Disease 2019 study concluded a significant increase in mental health disorders, from 80.8 million to 125.3 million between 1990 to 2019, causing significant disability and a negative impact on the economy and quality of life globally [2]. Recent evidence claims physical activity can have a positive effect on mental health, especially depression, stress, anxiety, and mood; however, convincing evidence exists only for children and adolescents [1,3]. Nevertheless, not all domains of physical activity—such as household, occupational, and travel—have been extensively researched for their impact, despite substantial accumulating evidence highlighting the favorable mental health benefits of leisure-time exercise (LTE) [4,5,6,7,8].

LTE refers to structured and planned physical activities performed outside occupational or household tasks for the purpose of improving physical or mental well-being. LTE includes activities such as aerobic exercise (e.g., running, cycling), resistance training (e.g., weightlifting, bodyweight exercises), yoga, and mind–body exercises, provided they are performed in a structured manner [9]. LTE has garnered significant attention for its potential to enhance mental health outcomes, including reducing anxiety, depression, and stress while improving mood and overall well-being [4,5,6,7,8,9]. LTE was associated with significantly lower oxidative stress and stress hormones, leading to lower depression, stress, and anxiety, and improvements in mood and well-being [10,11]. LTE mediates positive mental health by established key mechanisms: (1) stimulating mood, boosting neuro-transmitters such as serotonin, dopamine, and nor-epinephrine; (2) increasing production of Brain Derived Neurotrophic Factor (BDNF), a protein crucial for neuronal growth and plasticity; (3) regulating the pituitary–hypothalamus adrenal axis, which is responsible for the body’s stress response; (4) improving brain blood flow; and (5) promoting neuro-plasticity [12]. With the global rise in mental health challenges and the search for non-pharmacological interventions, understanding the breadth and depth of research on LTE’s mental health benefits is crucial. Randomized controlled trials (RCTs) provide the highest level of evidence in assessing the efficacy of interventions, yet a comprehensive understanding of the bibliometric characteristics of these studies remains unexplored.

This study aims to bridge this gap by providing a bibliometric analysis of RCTs included in a systematic review examining the mental health benefits of LTE among adults. Unlike systematic reviews, bibliometric analyses provide a broader, objective overview of publication trends, key contributions, and thematic developments in LTE’s impact on mental health. This approach serves as a valuable guide for future systematic reviews by highlighting high-impact studies, commonly used methodologies, and underrepresented areas in LTE research on mental health [13]. This analysis will highlight publication trends, influential authors, collaboration networks, research hotspots, and thematic shifts in the field, offering valuable insights for future research directions. The present bibliometric analysis study aimed to (1) evaluate the publication trends and global distribution of RCTs on LTE and mental health; (2) identify the most influential authors, institutions, and journals in this field; (3) map international collaborations and thematic research areas; and (4) explore research gaps and propose future directions for the study of LTE and mental health.

## 2. Materials and Methods

### 2.1. Screening Process

The articles were identified through a systematic review process conducted on 18 November 2024 using five databases, [i.e., PubMed, EMBASE, Web of Science, Ovid Medline, and the Cumulative Index for Nursing and Allied Health Literature]. Only RCTs evaluating the impact of LTE on mental health outcomes among adults were included. A standardized protocol guided the screening, data extraction, and quality assessment to ensure the reliability of the included studies. The primary author (KG) administered the search and downloaded the citations in .ris format. The downloaded citations were then exported to Endnote Web and two authors (KG and UG) independently screened the citations for the possible inclusion based on the eligibility criteria explained below. To strengthen the reliability of our study selection process, we have calculated the inter-rater agreement (Kappa index) between the two independent reviewers. Any discrepancies were resolved through mutual discussion and final agreement.

### 2.2. Eligibility Criteria

The eligibility criteria were based on Population, Intervention, Comparison, Outcomes, Study Design (PICOS) criteria. To be considered eligible for inclusion, the studies were required to (1) be randomized controlled trials, (2) involve adults (>18 years), (3) administer structured or planned exercise during leisure time (as defined by Caspersen et al. 1985 [9] as aerobic exercise (e.g., running, cycling), resistance training (e.g., weightlifting, bodyweight exercises), yoga, and mind–body exercises) for at least a week, (4) compare interventions with placebo or sham controlled interventions, and (5) measure any of the mental health outcomes (depression, anxiety, stress, mood, and well-being). Studies that were not published in full text, did not administer structured exercise during leisure time, administered interventions in youth (<18 years), or published in a language other than English were excluded.

### 2.3. Bibliometric Analysis

The potential citations were then downloaded in .txt format from Endnote Web and imported into SciVal, and the bibliometric data of the included studies were extracted. Key variables, including authorship, institutional affiliations, publication year, journal, keywords, and citation metrics, were analyzed. Bibliometric analysis was performed using Excel and SciVal to visualize: (1) co-authorship networks; (2) country and institutional collaborations; (3) keyword co-occurrence and thematic clusters; and (3) citation analysis to identify influential articles and journals. Field-weighted citation impact (FWCI), an indicator of the mean citation index, i.e. mean citations received by the article compared to the citations received by the documents of the same type (publication year and subject area), is used by research bodies and institutions to determine the relative research impact [14]. We established the FWCI from the SciVal websites. Besides SciVal, the text files were imported into the VOSviewer software version 1.6.20 and the documents were analyzed for bibliometric coupling for sources, citations, affiliations, and co-occurrence.

The following steps guided the bibliometric analysis: (1) Descriptive analysis: examined temporal publication trends, geographical distribution, and citation metrics; (2) Network analysis: evaluated co-authorship patterns, institutional linkages, and international collaborations; (3) Thematic mapping: performed keyword co-occurrence analysis to identify primary research themes and emerging areas; and (4) Citation analysis: identified high-impact journals, authors, and articles shaping the research landscape. The co-occurrence and bibliometric coupling were visualized using VOSviewer software, while a word cloud was retrieved from SciVal database.

## 3. Results

### 3.1. Screening

Of 5792 citations screened, 78 randomized controlled trials were deemed eligible for the systematic review. Cohen’s Kappa was found to be 0.64, suggesting moderate agreement between the two independent raters during the inclusion process. The majority of the citations were not relevant (n = 4691, 90%), and a few did not follow a protocol (n = 26, 0.5%), were not a systematic review (n = 18, 0.3%), did not include leisure-time exercise as an intervention (n = 142, 2.6%), were conducted in children (n = 186, 4%), or were not written in English or did not have a full text (n = 128, 2.5%) (Figure 1).

### 3.2. Bibliometric Analysis

#### 3.2.1. Descriptive Analysis

The included 78 randomized controlled trials had 452 authors, of which one-fourth of the studies (n = 24, 32%) had international collaboration. Almost half of the studies were published between 2017 and 2019 (Figure 2a); however, maximum citations were received during the period 2014–2015 (Figure 2b). All of the studies were from upper middle-income countries, except one (n = 1, 1.3%) conducted in India (Table 1, Figure 3). Most of the publications focused on psychological topics (Figure 4).

The majority of the included studies (n = 50, 64.1%) were in quartile one, and cumulative shares in Q1 and Q2 alone were significant (n = 68, 87%). University of Alberta (n = 4), National Taiwan University (n = 3), University of Melbourne (n = 3), and University of Western Australia (n = 3) were major contributors of the evidence that explored the effects of LTE on mental health through RCTs. Eighteen authors independently contributed a few (n = 2 each) to the underlying evidence (Table 2).

The highest number of studies were published in *International Journal of Environmental Research and Public Health* (n = 7, 9%), *PLOS One* (n = 5, 6.4%), and *Mental Health and Physical Activity* (n = 3, 4%), while others recorded a few citations only.

#### 3.2.2. Network Analysis

National collaboration was observed in the majority (n = 35, 44.9%) of the publications, followed by international collaboration (n = 24, 30.8%). Few publications resulted from within institutional collaboration (n = 17, 21.8%) and only one was a single-author publication [15]. Only a few studies (n = 2, 2.9%) had academic–industrial partnership. No patents or other intellectual property rights were identified in any of the studies. University of Alberta (n = 3, 4%) leads in international collaboration, followed by Curtin University, Universidad de São Paulo, University of British Columbia, University of Melbourne, and University of Western Australia, which have published two studies each with international collaborators.

#### 3.2.3. Thematic Mapping

The common key phrases identified in the word cloud were physical activity, exercise, kinesiotherapy, pedometer, fitness, actigraphy, heart rate variability, leisure activity, and mental health, while least addressed were patient compliance, sleep quality, mood, strength training. and mindfulness (Figure 5). However, when co-occurrences of the keywords were analyzed, stronger connections were found among physical activity, exercise, mood, well-being, and quality of life (Figure 6). Most of the trials questioned the validity of the mental health measures when administering leisure-time physical activity for mental health.

#### 3.2.4. Citation Analysis

A total of 2760 citations were received for 78 randomized controlled trials, with each trial receiving an average number of citations 35.4 per article. From 2014 until 2023, the average field-weighted citation impact was 1.50. Few studies (n = 14, 18%) were highly cited, while few (n = 24, 31%) were published in top 10% journals. Blumenthal et al. (1999) [16] remains an influential author, followed by Kubesch et al. (2003) [17], as depicted in Figure 7.

#### 3.2.5. Bibliographic Coupling

Bibliometric coupling measures the similarity of the documents using shared references (Blumenthal et al. (1991) [18], Blumenthal et al. (1999) [16], Kubesch et al. (2003) [17], Knubben (2007) [19], Krogh (2009) [20], Oertel-Knoechel (2014) [21]). Bergland, Blumenthal, Boen, Chen, Courneya, Cox, Gusi, Lan, and Ohta were found to be significant contributors (Figure 7). University of Western Australia, University of Melbourne, and University of Alberta contributed significantly (Figure 8). *International Journal of Environmental Research*; *Public Health*; *Mental Health and Physical Activity*; and *PLOS One* have published more than 30 papers in these areas.

## 4. Discussion

Findings from this bibliometric analysis provide valuable insight into the landscape of RCTs examining the effects of LTE on mental health among adults. Our bibliographic review highlights that existing evidence predominantly originates from high-income countries, particularly Australia, the United States, and the United Kingdom. This creates a research gap, hindering the generalization of mental health interventions in low- and middle-income countries. Additionally, the validity of mental health measures often remains a focal point of research. Our findings emphasize the deficiency in studies examining changes in individual mental health metrics, well-being, and quality of life. These findings not only contextualize the existing evidence base but also inform future research priorities to enhance the understanding and application of physical activity interventions for mental health.

Although a significant number of studies have explored the association of physical activity or exercise with mental health outcomes [22,23,24,25], only a fraction (1.4%) of the studies examined the leisure-time domain for mental health benefits. The temporal analysis of the RCTs revealed a significant number of studies published during 2017–2019, while there was a significant reduction in the studies during 2020, probably due to the COVID-19 pandemic and following lockdown, which induced a limitation on physical activity and exercise [26]. However, studies exploring the effect of LTE on mental health outcomes related to COVID-19 are on the rise [27]. Nonetheless, potential citations are decreasing, which requires attention. Almost two-thirds of the publications were in quartile 1, which demonstrates the importance of publishers showing interest on the effects of LTE on mental health outcomes. Key contributing countries included the United States, followed by Australia, Canada, and United Kingdom. The lesser evidence from the Asian countries warrants the need of quality trials to elucidate the effects of LTE on mental health outcomes [28]. The paucity of trials in the low- and middle-income countries demonstrates the lacunae in the awareness on the positive mental health benefits of LTE [29]. The most influential authors were identified to be from China and highly cited articles were published by the Fujian University of Traditional Chinese Medicine [30,31]. The maximum number of contributions was from the United States, followed by Australia, United Kingdom, and Canada. There was a paucity of trials exploring mental health and physical activity or exercise interventions in low- and middle-income countries.

Almost half of the included RCTs were published by international collaborators; however, the field-weighted citation index was high for nationally and institutionally collaborated publications. This depicts that heterogeneity in cultures, social norms in the practice of leisure-time activities, and mental health perceptions might have impacted the uniqueness of national and institutional collaborative publications [32]. University of Alberta leads the international collaborations in the theme of mental health outcomes with LTE, followed by Curtin University, Sao Paulo, British Columbia, University of Melbourne, and University of Western Australia. This demonstrates that Australian and Canadian universities are looking towards exploring the heterogeneity in cultural and social barriers to LTE in terms of mental health outcomes.

Mental health literacy, communication challenges, perceived social stigma, service limitations, lack of expertise, inadequate mental health training among healthcare providers, financial constraints, and discrimination are recognized as major barriers to addressing mental health issues [33]. Additionally, community and spiritual leaders play a significant role in shaping mental health perceptions due to socio-cultural taboos, gender disparities, joint family structures, and traditional beliefs [34]. A significant disparity exists in the availability of mental health professionals, such as psychiatrists, mental health nurses, and social workers, between high- and low-income countries [35]. In their narrative review, Rathod et al. (2017) emphasized that optimizing mental health services requires appropriate national resource allocation, the integration of tele-psychiatric counseling, and the training of alternative mental health providers [29]. This includes engaging community and spiritual leaders and promoting leisure-time physical activity as a key factor influencing mental well-being.

Although the effect of physical activity and exercise on mental health outcomes is a widely discussed topic, the least discussed themes in the included publications are patient compliance, sleep quality, strength training, and digital interventions to promote the effect of LTE on mental health outcomes. Patient compliance with behavioral interventions, especially LTE, is crucial for successful mental health outcomes [36]. Further, strength training has recently been found to promote positive mental health, but it is less discussed [37]. The use of digital interventions to promote physical activity have garnered interest because of their mental health benefits; however, there is a paucity of trials available to strengthen the existing evidence on the effect of digitally directed PA promotions on mental health outcomes [38,39]. Future trials exploring long-term effects, conducted in a multicentric manner and coordinated by international collaborators in low- and middle-income countries, are warranted. Additionally, systematic reviews exploring the effect of LTE on mental health outcomes with explicitly stated bias analysis and GRADE evidence synthesis are warranted.

### Limitations

The present bibliometric analysis has several limitations: (1) Ambiguity in the definition of leisure-time exercise (LTE): The lack of a standardized definition for LTE posed challenges for independent reviewers in determining study eligibility, potentially leading to inconsistencies in study inclusion; (2) Limitations of bibliometric analysis compared to systematic reviews: Unlike systematic reviews, bibliometric analysis does not assess the risk of bias, methodological quality, or intervention effectiveness of the included studies, limiting its ability to draw definitive conclusions about the evidence base; (3) Suboptimal validity of mental health measurement tools: The included studies used varying and, in some cases, suboptimal mental health assessment tools, making it difficult to ensure consistency in evaluating mental health outcomes; (4) Limited research from low- and middle-income countries: The scarcity of studies from LMICs reduces the global applicability and generalizability of the findings, as most research originates from high-income countries.

Despite these limitations, this bibliometric analysis provides valuable insights into publication trends, influential studies, and research gaps, serving as a foundation for future systematic reviews and meta-analyses in LTE and mental health research.

## 5. Conclusions

Significant growth has been observed in research exploring the impact of leisure-time exercise interventions on mental health issues such as stress, anxiety, and depression. However, the existing evidence predominantly originates from high-income countries. Additionally, the research landscape is characterized by limited collaboration, concerns regarding the validity of mental health measures, and a rise in digital interventions. Future trials should involve collaborations with low- and middle-income countries and utilize valid and reliable mental health outcome measurement tools. Further, future research directions should address additional measures of mental health, such as sleep quality, and examine the impact of strength training, digital interventions, and mindfulness.

## Figures and Tables

**Figure 1 healthcare-13-00575-f001:**
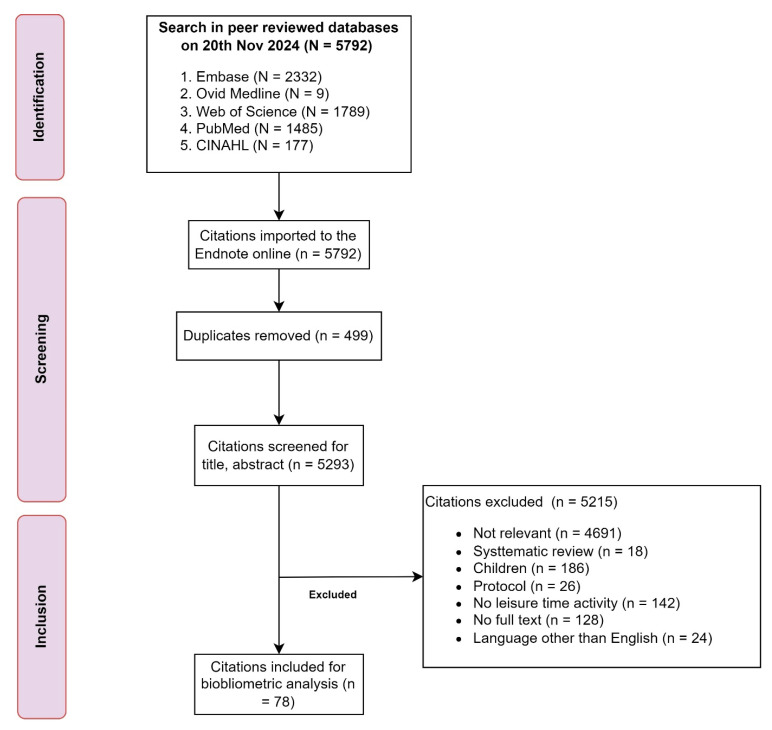
PRISMA flowchart guiding the screening and inclusion of studies for bibliometric analysis.

**Figure 2 healthcare-13-00575-f002:**
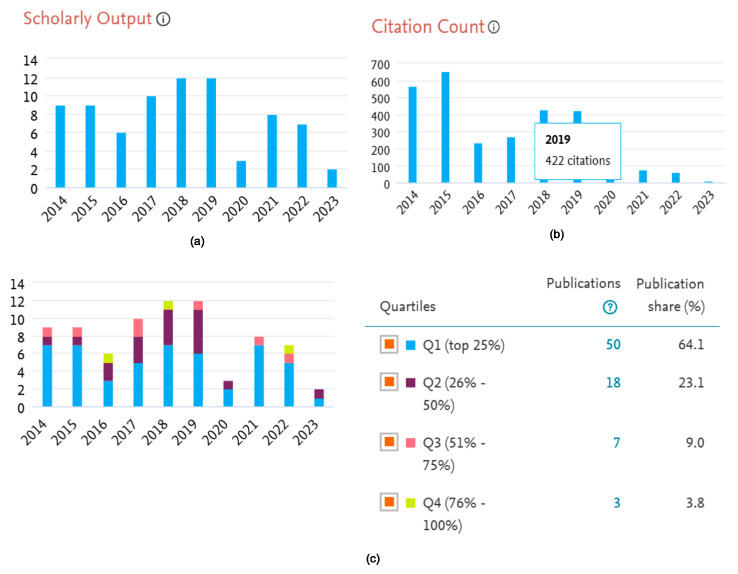
Descriptive analysis of the studies included for the systematic review: (**a**) shows year-wise number of publications; (**b**) shows citations across time; (**c**) shows the publications in different quartiles.

**Figure 3 healthcare-13-00575-f003:**
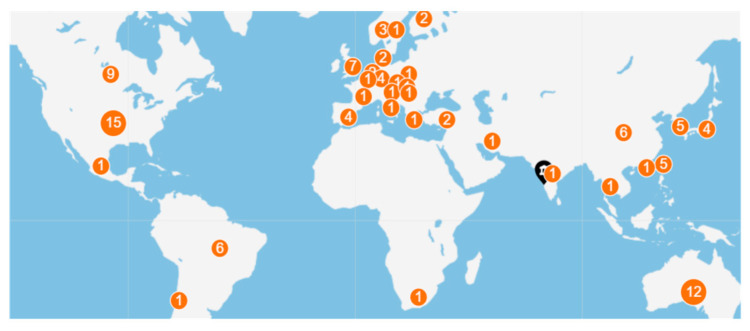
Global trends in the number of publications (numbers depicted). United States and Australia dominates in exploring the effect of leisure-time physical activity on mental health.

**Figure 4 healthcare-13-00575-f004:**
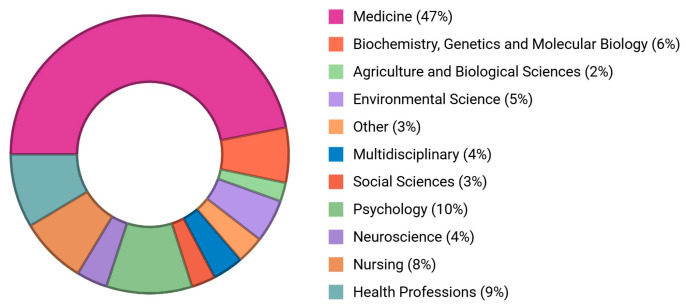
Pie chart shows the publication trends in different disciplines.

**Figure 5 healthcare-13-00575-f005:**
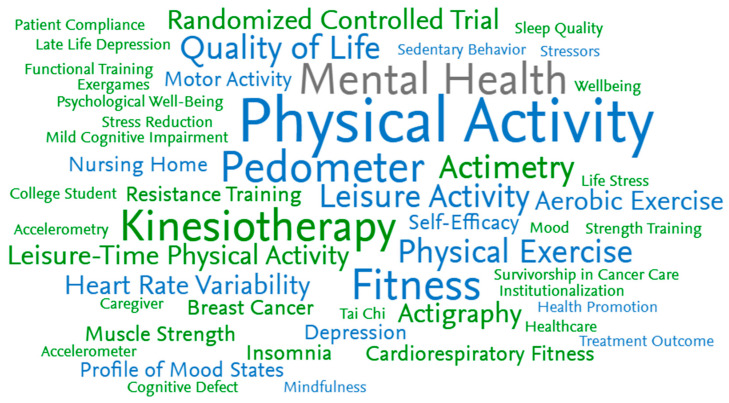
Top 50 key phrases identified in the included studies.

**Figure 6 healthcare-13-00575-f006:**
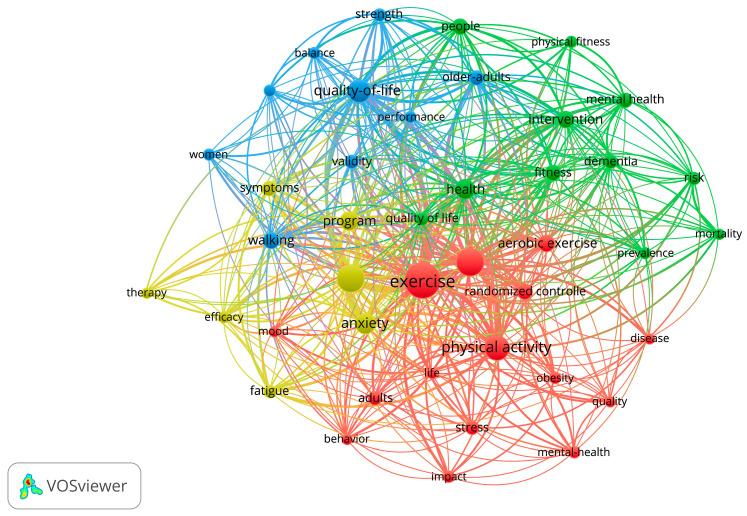
Co-occurrences of the keywords and the strength of interconnections.

**Figure 7 healthcare-13-00575-f007:**
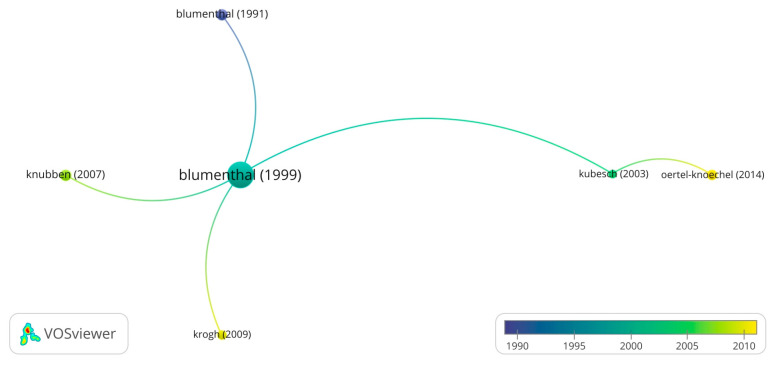
Highly cited authors and the connections [16,17,18,19,20,21].

**Figure 8 healthcare-13-00575-f008:**
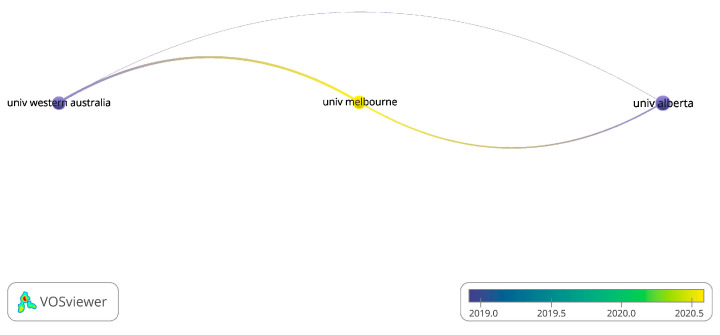
Affiliations of the most influential documents.

**Table 1 healthcare-13-00575-t001:** Countries where the included studies were conducted, number of studies, field-weighted citation, and number of citations.

Country/Region	Scholarly Output	Views	FWCI *	Citation
United States	15	947	1.66	519
Australia	12	672	1.58	342
Canada	9	471	1.62	281
United Kingdom	7	338	1.34	179
Brazil	6	363	2.49	295
China	6	311	0.95	166
South Korea	5	219	0.92	138
Taiwan	5	276	1.72	247
Germany	4	220	2.11	276
Japan	4	214	2.29	154
Spain	4	315	1.6	133
Norway	3	355	2.32	231
Denmark	2	253	2.9	120
Finland	2	70	0.3	24
Netherlands	2	242	1.26	62
Turkey	2	91	1.14	52
Austria	1	103	1.17	14
Belgium	1	64	0.33	4
Chile	1	51	1.73	31
Czech Republic	1	140	3.8	78
France	1	52	0.81	39
Greece	1	95	1.09	34
Hong Kong	1	19	1.51	6
Hungary	1	27	1.23	13
India	1	36	0.62	27
Iran	1	97	0.86	35
Italy	1	49	1.33	11
Mexico	1	29	0.23	2
Poland	1	20	0	0
Slovakia	1	103	1.17	14
South Africa	1	44	1.42	36
Sweden	1	22	1.04	14
Thailand	1	28	1.13	5

* FWCI—field-weighted citation index (ratio of the actual number of citations received by an output to date and the ‘expected’ number for an output with similar characteristics).

**Table 2 healthcare-13-00575-t002:** Influential authors who have maximum contributions exploring the impact of leisure time exercise on mental health.

Author	Affiliation	Country/Region	FWCI
Bergland, Astrid	Oslo Metropolitan University	Norway	1.98
Chen, Bai	Fujian University of Traditional Chinese Medicine	China	1.19
Chen, Lidian	Fujian University of Traditional Chinese Medicine	China	1.19
Courneya, Kerry S.	University of Alberta	Canada	0.74
Cox, Kay L.	University of Western Australia	Australia	1.44
Engedal, Knut A.	University of Oslo	Norway	1.98
Fang, Qianying	Fujian University of Traditional Chinese Medicine	China	1.19
Huang, Hanchung	China University of Technology	Taiwan	1.37
Lan, Xiulu	Fujian University of Traditional Chinese Medicine	China	1.19
Lautenschlager, Nicola T.	Royal Melbourne Hospital	Australia	1.44
Li, Junzhe	Fujian University of Traditional Chinese Medicine	China	1.19
Li, Moyi	Nanchang University	China	1.19
McAuley, Edward D.	University of Illinois at Urbana-Champaign	United States	0.55
Tao, Jing	Fujian University of Traditional Chinese Medicine	China	1.19
Telenius, Elisabeth Wiken	Vestfold Hospital Trust	Norway	1.98
Teng, Ching I.	Chang Gung Memorial Hospital	Taiwan	1.37
Zheng, Guohua	Shanghai University of Traditional Chinese Medicine	China	1.19
Zheng, Xin	Fujian University of Traditional Chinese Medicine	China	1.19

FWCI—field-weighted citation index.

## Data Availability

The data used in this bibliometric analysis were extracted from publicly available, published studies. All data sources are properly referenced in the manuscript. Additional information or datasets can be obtained from the corresponding author upon reasonable request.

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
