# Peer review of "The Effect of Leisure-Time Exercise on Mental Health Among Adults: A Bibliometric Analysis of Randomized Controlled Trials"

_healthcare, 2025, doi:10.3390/healthcare13050575_

Round 1
Reviewer 1 Report
Comments and Suggestions for Authors
In bibliographic matching, cluster analysis and graphics should be given about the similarity of the articles to each other in terms of subject. It is quite difficult for the reader to establish a connection between the figure given in the article and those mentioned in the discussion section.
Additional comments
Network analyses of the 50 most important keywords identified in the included studies and graphs of bibliometric networks based on database searches using VOSviewer are acceptable if included in the text. Sample graphs are provided in the attached file.

Author Response
We thank the reviewers for their positive and valuable feedback on our manuscript entitled “Leisure time exercise on mental health among adults: A bibliometric analysis of randomized controlled trials”. We think that the reviewers’ comments substantially assisted in improving the content and clarity of the paper. Outlined below, you will find point-by-point responses to each of the comments raised by the reviewers. Changes were highlighted in yellow in the revised version of the manuscript.
Authors’ response to the comments of Reviewer # 1
- Reviewer’s comment: In bibliographic matching, cluster analysis and graphics should be given about the similarity of the articles to each other in terms of subject. It is quite difficult for the reader to establish a connection between the figure given in the article and those mentioned in the discussion section.
Authors’ response: Thank you for your legitimate comment. As suggested by the reviewer, we have re-computed a cluster analysis using VOSviewer.
Changes made: lines 185 - 205
- Reviewer’s comment: Network analyses of the 50 most important keywords identified in the included studies and graphs of bibliometric networks based on database searches using VOS viewer are acceptable if included in the text. Sample graphs are provided in the attached file.
Authors’ response: As mentioned previously, we have created the respective figures and included those in the text.
Changes made: lines 185 - 205

Reviewer 2 Report
Comments and Suggestions for Authors
The topic is very pertinent and current, so it has a place in the journal.
The summary should obtain the databases that have been used. The summary also contains typos such as repeated words.
The key words should be ordered alphabetically.
The review period is also not clear.
They should make the inclusion and exclusion criteria more clear and add quality criteria by enhancing the Kappa index.
In the discussion, they should work in more depth on the most impactful articles on the subject.
The conclusions are very enlightening, but it is necessary to add future lines of research based on the literature reviewed.
Author Response
We thank the reviewers for their positive and valuable feedback on our manuscript entitled “Leisure time exercise on mental health among adults: A bibliometric analysis of randomized controlled trials”. We think that the reviewers’ comments substantially assisted in improving the content and clarity of the paper. Outlined below, you will find point-by-point responses to each of the comments raised by the reviewers. Changes were highlighted in yellow in the revised version of the manuscript.
Authors’ response to the comments of Reviewer # 2
- Reviewer’s comment: The topic is very pertinent and current, so it has a place in the journal.
Authors’ response: We thank the reviewer for his/her overall positive rating of our manuscript.
- Reviewer’s comment: The summary should obtain the databases that have been used. The summary also contains typos such as repeated words.
Authors’ response: We have revised the abstract and changed grammatical, syntactical errors whenever needed.
Changes made: whole abstract
- Reviewer’s comment: The key words should be ordered alphabetically.
Authors’ response: The keywords were ordered alphabetically as requested by the reviewer.
Changes made: line 42
- Reviewer’s comment: The review period is also not clear.
Authors’ response: We have revised the search period in the abstract:
“Five electronic databases were searched from their inception until 20th November 2024.”
Changes made: lines 25, 26
- Reviewer’s comment: They should make the inclusion and exclusion criteria more clear and add quality criteria by enhancing the Kappa index.
Authors’ response: We have modified the eligibility criteria as suggested by the reviewer. The inclusion and exclusion criteria should now be clearer to the readers. The Kappa index for the agreement between two independent reviewers amounted to 0.64. Changes were made in the results section (lines 90-92, 95 – 102, 128 – 129).
- Reviewer’s comment: In the discussion, they should work in more depth on the most impactful articles on the subject.
Authors’ response: Thank you for your legitimate comment. We have followed your suggestion and revised the discussion section.
Changes made: lines 215-222
- The conclusions are very enlightening, but it is necessary to add future lines of research based on the literature reviewed.
Authors’ response: Thank you for your affirmative comment on our discussion section. In accordance with your suggestion, we have revised the conclusions
“An exponential growth has occurred over the past years with regards to the published studies that examined the effects of leisure-time physical activity interventions on mental health issues such as stress, anxiety, and depression. However, the existing evidence predominantly originates from high-income countries. Additionally, methodological limitations exist with regards to the validity of mental health measures. To receive a more conclusive picture on this topic, future research should primarily look at the impact of leisure-time physical activity programs on mental health in low- and middle-income countries and utilize valid and reliable mental health outcome measures.”
Changes made: 270 - 277

Reviewer 3 Report
Comments and Suggestions for Authors
Thank you for submitting to Healthcare.
Please see the attachment.

Author Response
We thank the reviewers for their positive and valuable feedback on our manuscript entitled “Leisure time exercise on mental health among adults: A bibliometric analysis of randomized controlled trials”. We think that the reviewers’ comments substantially assisted in improving the content and clarity of the paper. Outlined below, you will find point-by-point responses to each of the comments raised by the reviewers. Changes were highlighted in yellow in the revised version of the manuscript.
Authors’ response to the comments of Reviewer # 3
- Reviewer’s comment: Major comment - Although this document is a bibliometric analysis, it has many limitations for publication. Many bibliometric analyses related to health generally assume a scientific analysis approach through annual trends, changes in major interests, or formulas, calculation indexes, etc. Compared to other bibliometric studies, the author's study is too simple and does not clearly provide any information for health management. We believe that it would be very valuable if the authors could bibliographically analyze past research trends in healthcare and conduct modeling studies that could predict future trends
Authors’ response: Thank you for you having raised this important issue. Previously, researchers from observational studies reported significant positive associations between leisure-time, physical activity and potential health benefits. In converse this favorable association was not established with other domains of physical activity such as household, travel or occupational physical activity. This led to the raise of significant randomized trials in recent years to explore the effectiveness of the leisure time physical activity on health benefits especially mental health among adults.
- Reviewer’s comment: Divide Figure 2 into 3 (above map, below map, below map) and increase the resolution.
Authors’ response: We have divided the figures into 3 sections as suggested by the reviewer and increased resolution to 300 dpi.
Changes made: Figures 2, 3 and 4
- Reviewer’s comment: There is only the date of search. Please indicate the period included in the search criteria.
Authors’ response: The keywords were ordered alphabetically. Thank you.
Changes made: line 42
- Reviewer’s comment: Number the research method and results. 2.1/ 2.2. / ...3.1 / 3.2 / ...
Authors’ response: We have revised the results section according to the reviewer’s comment.
Changes made: lines 25, 26
- Reviewer’s comment: Analysis: Did the author use statistics? How were % and FWCI calculated? Please write in detail so that readers can understand.
Authors’ response: Please note that we have retrieved the % and FWCI from the SciVal databases. A new paragraph was included to better explain the FWCI to the readers:
The “Field Weighted Citation Impact (FWCI) is an indicator of the mean number of citations an article has received compared to the number of citations similar articles with regards to publication year and subject area have received. This FWCI has frequently been used by research bodies and institutions to rate the relative research impact of scientific articles (Purkayastha et al., 2019). We have used the reported FWCI from the SciVal website. Besides the application of SciVal, the text files of the included articles were imported to the VOSviewer software version 1.6.20 and the documents were analyzed for bibliometric coupling of the underlying sources, citations, affiliations and co-occurrence analysis”. The respective information has been added to the methodology section for better transparency of the applied methods and to replicate the methodological approach of our study in future research.
Changes made: lines 110 – 117
- Reviewer’s comment: FWCI is understood to be one of the important contents. However, there is no information about FWCI in the research method and discussion.
Authors’ response: We have added a new paragraph to the methods section where we describe and explain the FWCI.
Changes made: lines 110 – 117
- Reviewer’s comment: What program did you use for the word cloud? And write the process in detail in the research method.
Authors’ response: We have retrieved the word clouds from Scival. We have added this information to the methods section:
“[…] while word cloud was retrieved from SciVal database.”
Changes made: line 125

Reviewer 4 Report
Comments and Suggestions for Authors
This study presents valuable insights into the research trends on LTE and mental health through bibliometric analysis. However, addressing the above suggestions—such as clarifying the definition of LTE, explicitly stating the limitations of bibliometric analysis, discussing cultural and social influences in more detail, and proposing concrete policy recommendations—will significantly enhance the study's depth, clarity, and impact.
**Introduction
- It would be beneficial to include neuroscientific or psychological mechanisms that explain the relationship between exercise and mental health. Providing such background will strengthen the theoretical foundation of the study.
- The study mentions that cultural and social factors influence the relationship between exercise and mental health, but it lacks a clear explanation of which specific cultural factors are involved. For instance, does it refer to differences in the cultural definition of leisure-time exercise or varying cultural perceptions of mental health? A more detailed discussion on this aspect would enhance clarity.
- The study should more explicitly highlight how it differs from previous research. Since numerous studies have already explored the impact of exercise on mental health, emphasizing this study’s unique contribution will be crucial. For example, if the goal is to analyze the publication trends and research network of LTE and mental health studies, it would be helpful to clearly define bibliometric analysis at the beginning.
- The purpose of bibliometric analysis needs to be explicitly clarified. Since this method does not include a qualitative assessment, its limitations should be communicated to readers to ensure a comprehensive understanding.
**Materials and Methods
- The definition of structured/planned leisure-time exercise (LTE) is not clearly provided. It is unclear what types of exercises are included (e.g., aerobic exercise, strength training, yoga, etc.). Clarifying the LTE definition and specifying the types of exercises considered would enhance the study's transparency.
- The exclusion criteria require further justification. The study excludes children and adolescents, but the effect of exercise on mental health may vary by age group. It is important to explain why this particular age group was excluded.
- The limitations of bibliometric analysis should be explicitly stated. While this method is useful for understanding research trends, it does not assess the quality of individual studies. This distinction should be emphasized both in the methods section and in the study's discussion of limitations.
- Additionally, the differences between bibliometric analysis and systematic review should be clearly outlined. Readers should understand why this study opted for bibliometric analysis instead of a systematic review and what implications this choice has for the interpretation of findings.
**Results
- The study should discuss why research from low- and middle-income countries (LMICs) is limited and suggest policy recommendations to address this issue. This would add depth to the findings and provide a more comprehensive perspective.
- Future research directions should include additional topics closely related to mental health, such as sleep quality, strength training, digital interventions, and mindfulness. These areas are relevant and could significantly contribute to the field of LTE and mental health studies.
**Discussion
- The study mentions the regional concentration of research in Western countries, but it does not propose specific solutions to encourage research in LMICs. The authors could provide recommendations for how to increase research output in underrepresented regions, such as through funding initiatives, international collaborations, or policy interventions.
- Future studies should incorporate qualitative assessment tools such as the GRADE system or Risk of Bias assessment to evaluate the quality of the included studies. This would strengthen the reliability of future research in this area.
Author Response
We thank the reviewers for their positive and valuable feedback on our manuscript entitled “Leisure time exercise on mental health among adults: A bibliometric analysis of randomized controlled trials”. We think that the reviewers’ comments substantially assisted in improving the content and clarity of the paper. Outlined below, you will find point-by-point responses to each of the comments raised by the reviewers. Changes were highlighted in yellow in the revised version of the manuscript.
Authors’ response to the comments of Reviewer # 4
- Reviewer’s comment: This study presents valuable insights into the research trends on LTE and mental health through bibliometric analysis. However, addressing the above suggestions—such as clarifying the definition of LTE, explicitly stating the limitations of bibliometric analysis, discussing cultural and social influences in more detail, and proposing concrete policy recommendations—will significantly enhance the study's depth, clarity, and impact.
Authors’ response: Thank you for your helpful comments which certainly contribute to enhance the quality of this paper. We have now included a detailed analysis of the publication trends, co-occurrences, citation analyses, and bibliographic coupling. Further, we have revised the methods sections and included additional information on the usage of VOSviewer. Moreover, we have added additional figures to better visualize our results (figures 6 – 8). The discussion and conclusion sections were re-written in large parts and included a section on study limitations and recommendations for future research avenues.
Change: throughout the manuscript
- Reviewer’s comment: **Introduction: It would be beneficial to include neuroscientific or psychological mechanisms that explain the relationship between exercise and mental health. Providing such background will strengthen the theoretical foundation of the study.
Authors’ response: We agree with the reviewer’s comment and added information on neurobiological mechanisms relating the exercise and mental health:
“LTE mediates mental health through the following neurobiological mechanisms reported in the scientific literature (Linnenluecke et al., 2019):
(1) neurotransmitters such as serotonin, dopamine and nor-epinephrine stimulate individual’s mood: (2) brain derived neurotrophic factor (BDNF), a protein crucial for neuronal growth and plasticity, enhances; (3) the pituitary-hypothalamus adrenal axis regulates individual’s stress responses; (4) improving brain blood flow and (5) promoting neuro plasticity (Ren & Xiao, 2023).”
- Reviewer’s comment: **Introduction: The study mentions that cultural and social factors influence the relationship between exercise and mental health, but it lacks a clear explanation of which specific cultural factors are involved. For instance, does it refer to differences in the cultural definition of leisure-time exercise or varying cultural perceptions of mental health? A more detailed discussion on this aspect would enhance clarity.
Authors’ response: We agree with the reviewer’s comment. Thank you. As the study objective was to identify the available evidence on the effects of LTE on mental health, we have revised different sections of the manuscript and included for instance more detailed information on the study rationale with regards to how LTE may impact or mediate mental health. We have additional included information in the discussion section on the socio-cultural barriers of LTE for mental health in low-middle income countries,
“Mental health literacy, communication challenges, perceived social stigma, service limitations, and lack of expertise, inadequate mental health training among healthcare providers, financial constraints, and discrimination are recognized as major barriers to address mental health issues (Krystallidou et al., 2024). Additionally, community and spiritual leaders play a significant role in shaping mental health perceptions due to socio-cultural taboos such as gender disparities, joint family structures, and traditional beliefs (Mboweni et al., 2024). A significant disparity exists in the availability of mental health professionals, such as psychiatrists, mental health nurses, and social workers, between high- and low-income countries (Bruckner et al., 2011). In their narrative review, Rathod et al. (2017) emphasized that optimizing mental health services requires appropriate national resource allocation, the integration of telepsychiatric counseling, and the training of alternative mental health providers (Rathod et al., 2017). This includes engaging community and spiritual leaders and promoting leisure-time physical activity as a key factor influencing mental well-being.”
Changes made: lines 300 - 316
- Reviewer’s comment: **Introduction: The study should more explicitly highlight how it differs from previous research. Since numerous studies have already explored the impact of exercise on mental health, emphasizing this study’s unique contribution will be crucial. For example, if the goal is to analyze the publication trends and research network of LTE and mental health studies, it would be helpful to clearly define bibliometric analysis at the beginning.
Authors’ response: Thank you for your comment. We have included additional information in the revised version of this manuscript where we explain why we conducted a bibliometric analysis and not a systematic literature review:
“Unlike systematic literature reviews, bibliometric analyses provide a broader overview of publication trends, key contributions, and thematic developments with regards to the impact of LTE on mental health. The bibliometric analysis is a precursor of subsequent systematic literature reviews on the topic by by highlighting high-impact studies, commonly used methodologies, and underrepresented areas in LTE research on mental health (Linnenluecke et al., 2019).”
Changes made: lines 82 – 86
- Reviewer’s comment: **Introduction: The purpose of bibliometric analysis needs to be explicitly clarified. Since this method does not include a qualitative assessment, its limitations should be communicated to readers to ensure a comprehensive understanding.
Authors’ response: Please see our answer to the previous comment.
Changes made: lines 82 – 86
- Reviewer’s comment: **Materials and Methods: The definition of structured/planned leisure-time exercise (LTE) is not clearly provided. It is unclear what types of exercises are included (e.g., aerobic exercise, strength training, yoga, etc.). Clarifying the LTE definition and specifying the types of exercises considered would enhance the study's transparency.
Authors’ response: We have added a LTE definition to the introduction section as requested by the reviewer.
“LTE includes structured and planned physical activity performed outside occupational or household tasks for the purpose of improving physical or mental well-being. As such, LTE comprises activities such as aerobic exercise (e.g., running, cycling), resistance training (e.g., weightlifting, strength training), yoga, and mind-body exercises, provided they are performed in a structured manner (Caspersen et al., 1985).”
Further we have added a third inclusion criteria in the methods section:
“(3) administered structured or planned exercise during leisure time (as defined by Cas-persen et al. 1985 as aerobic exercise (e.g., running, cycling), resistance training (e.g., weightlifting, body-weight exercises), yoga, and mind-body exercises for a duration ≥ one week.”
Changes made: lines 59 – 62; 111 - 113
- Reviewer’s comment: **Materials and Methods: The exclusion criteria require further justification. The study excludes children and adolescents, but the effect of exercise on mental health may vary by age group. It is important to explain why this particular age group was excluded.
Authors’ response: Thank you. We specifically aimed to provide a bibliographic analysis on LTE effects on mental health in adults. Accordingly, we have excluded children and adolescents. Mental health problems of children and adolescents are significantly different from those of adults (family role, financial problems, work stress). We agree that the topic is relevant in youth as well but this would be another bibliographic analysis.
- Reviewer’s comment: **Materials and Methods: The limitations of bibliometric analysis should be explicitly stated. While this method is useful for understanding research trends, it does not assess the quality of individual studies. This distinction should be emphasized both in the methods section and in the study's discussion of limitations.
Authors’ response: We have addressed this as a new subheading in the discussion section
“4.1. Study limitations
This bibliometric analysis has several methodological limitations that need to be acknowledged: (1) the term LTE is not clearly defined in the literature which is why it was difficult to determine study eligibility, potentially leading to inconsistencies in study selected and inclusion. In case of discrepancy between two raters (K.G, U.G), a third rater (K.P) was contacted and ultimately a unanimous decision was found whether the study was included or exluded; (2) bibliometric analyses compared with systematic reviews do not assess the risk of bias, methodological quality, or intervention effectiveness of the included studies, limiting its ability to draw definitive conclusions about the evidence base; (3) the identified mental health tests are limited with regards to their validity making it difficult to ensure consistency in evaluating mental health outcomes; (4) the identified evidence primarily comes from high-income countries. Hardly any information is available on LTE effects on mental health in low- and middle-income countries: The scarcity of studies from low income countries prevents transferability of our findings to these countries.
Despite these study limitations, this bibliometric analysis provides valuable insights into publication trends, impactful studies, and research gaps, serving as a foundation for future systematic reviews and meta-analyses on the LTE effects on mental health.”
Changes made: lines 300 - 316
- Reviewer’s comment: **Materials and Methods: Additionally, the differences between bibliometric analysis and systematic review should be clearly outlined. Readers should understand why this study opted for bibliometric analysis instead of a systematic review and what implications this choice has for the interpretation of findings.
Authors’ response: We agree – thank you. We have added a statement to the introduction section to better explain why we used a bibliometric analysis
“Unlike systematic reviews, bibliometric analysis provides a broader, objective overview of publication trends, key contributions, and thematic developments in LTE's impact on mental health. This approach serves as a valuable guide for future systematic reviews by highlighting high-impact studies, commonly used methodologies, and underrepre-sented areas in LTE research on mental health (Linnenluecke et al., 2019).”
Changes made: lines 82 – 86
- Reviewer’s comment: **Results: The study should discuss why research from low- and middle-income countries (LMICs) is limited and suggest policy recommendations to address this issue. This would add depth to the findings and provide a more comprehensive perspective.
Authors’ response: Thank you. We have added a new paragraph to the discussion section to address the raised query made by the reviewer:
“Mental health literacy, communication challenges, perceived social stigma, service limitations, lack of expertise, inadequate mental health training among healthcare providers, financial constraints, and discrimination are recognized as major barriers to ad-dressing mental health issues (Krystallidou et al., 2024). Additionally, community and spiritual leaders play a significant role in shaping mental health perceptions due to so-cio-cultural taboos, gender disparities, joint family structures, and traditional beliefs (Mboweni et al., 2024). A significant disparity exists in the availability of mental health professionals, such as psychiatrists, mental health nurses, and social workers, between high- and low-income countries (Bruckner et al., 2011). In their narrative review, Rathod et al. (2017) emphasized that optimizing mental health services requires appropriate national resource allocation, the integration of telepsychiatric counseling, and the training of alternative mental health providers (Rathod et al., 2017). This includes en-gaging community and spiritual leaders and promoting leisure-time physical activity as a key factor influencing mental well-being.”
Change: 274 – 286 lines
- Reviewer’s comment: **Results: Future research directions should include additional topics closely related to mental health, such as sleep quality, strength training, digital interventions, and mindfulness. These areas are relevant and could significantly contribute to the field of LTE and mental health studies.
Authors’ response: We thank the reviewer for the positive rating.
We have further revised the conclusions to provide future research avenues.
- Reviewer’s comment: **Discussion: The study mentions the regional concentration of research in Western countries, but it does not propose specific solutions to encourage research in LMICs. The authors could provide recommendations for how to increase research output in underrepresented regions, such as through funding initiatives, international collaborations, or policy interventions.
Authors’ response: We agree with the reviewer regarding the recommendations for increasing the research output in under represented areas. We have revised the respective paragraph in the discussion as follows:
“In their narrative review, Rathod et al. (2017) emphasized that optimizing mental health services requires appropriate national resource allocation, the integration of telepsychiatric counseling, and the training of alternative mental health providers (Rathod et al., 2017). This includes engaging community and spiritual leaders and promoting leisure-time physical activity as a key factor influencing mental well-being”
Changes made: lines 282- 286
- Reviewer’s comment: **Discussion: Future studies should incorporate qualitative assessment tools such as the GRADE system or Risk of Bias assessment to evaluate the quality of the included studies. This would strengthen the reliability of future research in this area.
Authors’ response: We have added the following statement to the discussion section:
“Nevertheless, systematic reviews exploring the effects of LTE on mental health outcomes with explicitly stated bias analysis and GRADE evidence synthesis are warranted.”
Changes made: lines 268 - 269

Round 2
Reviewer 1 Report
Comments and Suggestions for Authors
The corrections and additional graphics made by the author are appropriate and sufficient. Acceptable.
Reviewer 3 Report
Comments and Suggestions for Authors
I do not have any comment.